# Interstitial Lung Diseases in Israel: Large Variability in Close Geographic Proximity

**DOI:** 10.3390/diagnostics15212780

**Published:** 2025-11-02

**Authors:** Tzlil Hershko, Ophir Freund, Sonia Schneer, Michael J. Segel, Ori Wand, Amir Bar-Shai, David Shitrit, Liran Levy, Yochai Adir, Avraham Unterman

**Affiliations:** 1Institute of Pulmonary Medicine, Tel Aviv Sourasky Medical Center, Gray Faculty of Medical and Health Sciences, Tel Aviv University, Tel Aviv 6423906, Israelophir068@gmail.com (O.F.);; 2Center of Excellence for Interstitial Lung Diseases, Tel Aviv Sourasky Medical Center, Gray Faculty of Medical and Health Sciences, Tel Aviv University, Tel Aviv 6423906, Israel; 3Pulmonary Division, Lady Davis Carmel Medical Center, Faculty of Medicine, The Technion Institute of Technology, Haifa 3436212, Israel; 4Pulmonary Institute, Sheba Medical Center, Gray Faculty of Medical and Health Sciences, Tel Aviv University, Ramat Gan 5266202, Israel; 5Pulmonary Department, Meir Medical Center, Gray Faculty of Medical and Health Sciences, Tel Aviv University, Kfar Saba 4428164, Israel; 6Division of Pulmonary Medicine, Barzilai University Medical Center, Faculty of Health Sciences, Ben-Gurion University of the Negev, Ashkelon 7830604, Israel

**Keywords:** pulmonary fibrosis, interstitial lung diseases, idiopathic pulmonary fibrosis, registry, diagnosis

## Abstract

**Background:** There have been no previous comprehensive reports on interstitial lung diseases (ILD) in Israeli population, that may have unique epidemiological features. We aimed to explore ILD in Israel, with an emphasis on disparities between different regions of the country. **Methods:** The study included consecutive patients with a multidisciplinary diagnosis of ILD, using data from registries of four tertiary medical centers (MC) located in Central and Northern Israel. Multivariate regression models were used to assess the region of residence (peripheral vs. central) as an independent predictor for ILD subtypes. **Results:** Included were 927 patients with ILD (mean age 67 ± 13, 40% females). Most patients (56–61%) reported working in at least one job that involved relevant inhalational exposures. Despite the geographic proximity of MCs (all within 100 km), significant variations in demographic and clinical characteristics were observed, including age, sex, exposures, and ILD diagnoses (*p* < 0.01). The most prevalent diagnoses were Idiopathic pulmonary fibrosis (IPF, range 13–58%) and autoimmune-related ILD (11–30%). In peripheral areas, the diagnosis of IPF was more frequent (53% vs. 24%, *p* < 0.01), while exposure-related ILD (5% vs. 16%, *p* < 0.01) and autoimmune-related ILD (16% vs. 25%, *p* < 0.01) were more frequent in central Israel. In multivariate analysis, peripheral residence remained an independent predictor for IPF (AOR 2.95, 95% CI 2.1–4.1) and central residence for exposure-related ILD (AOR 0.46, 95% CI 0.33–0.63). **Conclusions:** Variations in ILD characteristics were observed between centers in close geographic proximity, highlighting disparities between peripheral and central Israel, and the need for personalized assessment based on local frequencies and exposures.

## 1. Introduction

Interstitial lung disease (ILD) is a broad term encompassing over 200 fibrotic and non-fibrotic disorders that diffusely affect lung parenchyma. Typically, ILD causes lung restriction and impaired gas exchange, resulting in breathlessness, diminished exercise tolerance, reduced quality of life, and in some cases, shortened survival [1]. Diagnosis of a specific ILD relies on the combination of clinical, radiological, and pathological characteristics, preferably through a multidisciplinary discussion (MDD) [2,3,4].

The clinical course, therapeutic strategy, and prognosis depend on the underlying cause of the ILD and whether it is fibrotic or not [5,6]. The distribution of ILD etiologies varies between countries and regions, as it is affected by local demographics and exposures [3,7]. Therefore, it is imperative for pulmonologists to know the local ILD characteristics in their region.

Although many local and regional ILD registries were established over the last decade [8,9], there have been no comprehensive reports regarding ILDs in the Israeli population that may have unique demographic and ethnic characteristics. The aim of this study was to explore the characteristics of ILDs in Israel, with a focus on differences between the periphery and the center of Israel to identify potential disparities.

## 2. Materials and Methods

This is an observational multicenter study including consecutive patients with ILD. Data were obtained from ILD registries of four large tertiary medical centers in Israel: Tel Aviv Medical Center (TA-MC), Carmel MC (C-MC), Meir MC (M-MC), and Sheba MC (S-MC). All registries were retrospective, except that of TA-MC which was prospective and included consecutive ILD patients who signed an informed consent form and were recruited between May 2021 and August 2022. Dates of first clinic visit for the retrospective registries were in the range of 2011–2018, 2015–2021, 2011–2022 for C-MC, M-MC, and S-MC, respectively. All included patients were adults (above 18 years old). ILD diagnoses were assigned by MDD according to guidelines available at the time of diagnosis and as described before [10,11,12]. In general, the MDD included pulmonologists, radiologists, rheumatologists (on per-case basis), and other specialists [13]. Patients with non-ILD diagnoses were excluded. Sarcoidosis patients were included provided they had interstitial lung involvement, except for C-MC and M-MC registry which did not include sarcoidosis patients.

The study was conducted according to the principles of the World Medical Association Declaration of Helsinki and GCP Guidelines and approved by the institutional review board (0101-21-TLV approval date: 13 April 2021 and 0456-21-TLV approval date: 7 July 2021).

### 2.1. Data Collection

The four registries contained demographic and clinical data that were extracted from electronic patient records, including pulmonology clinic visits, hospitalizations, or visits with other specialists. Demographic data included age, sex, country of birth, place of residence, ethnicity, and occupation. Clinical data included smoking status, occupational and environmental exposures, co-morbid diseases, pulmonary function tests, treatment, and MDD diagnosis. These data were available for all four centers unless otherwise specified. As for TA-MC, data were also extracted from the Chest Interstitial and Diffuse Lung Disease Patient Questionnaire that was prospectively filled out by patients [14], which includes smoking history, exposures, occupational history, and medication history. Data were extracted for all databases by reviewing patients’ clinic visits, MDD diagnosis, and additional procedures made (bronchoscopy results, serological analysis, etc.). All four databases were constructed similarly, with mandatory fields based on the clinical and demographic variables described above. The datasets from each center were assessed for integrity, and variables were aligned into a common format before merging into the final database.

### 2.2. Data Analysis

Data are first presented separately for each of the four registries, and subsequently combined for statistical analysis. Given that patients with sarcoidosis were not included in two of the four registries, we did not include sarcoidosis in our pooled database for statistical analysis. Among the demographic variables, place of living was missing for patients from M-MC. Therefore, this center was excluded from analyses on peripheral vs. central areas. In addition, among clinical variables, only pulmonary function results had missing data (FVC—95 subjects, for DLCO—193 subjects), and these patients were excluded from relevant analyses. Continuous variables are presented as means (±standard deviation) for a normal distribution and as medians (inter-quartile range) for non-normal distribution.

Analysis of variance (ANOVA) or Kruskal–Wallis test were employed to compare normally and non-normally distributed continuous variables across all centers, respectively. Categorical variables were compared between the centers using Chi-squared test. The associations between variables and ILD diagnosis in central and peripheral Israel were assessed using univariate analysis. For this univariate analysis, we defined an exposure-related ILD category, which included all cases of hypersensitivity pneumonitis (HP), together with cases of pneumoconiosis (asbestosis, silicosis, etc.). Multivariate regression models were used to assess the place of residence as an independent predictor for selected ILDs, with other significant predictors based on the univariate analysis included in each model. *p*-values under 0.05 were considered statistically significant. All analyses were performed using SPSS version 28.0.

## 3. Results

Nine hundred and twenty-seven patients were included in this study, 321 from C-MC, 274 from S-MC, 200 from TA-MC and 132 from M-MC. S-MC, TA-MC and M-MC are located in the center of Israel, while C-MC is located in the north and considered peripheral. Figure 1 portrays the geographic distribution of subjects in each center, according to district. Subjects living in Central Israel (Dan, Center, and Jerusalem districts) comprised 94% and 77% of patients at TA-MC and S-MC, while all subjects from C-MC lived in the Haifa & Northern peripheral districts. No data was available regarding the distribution of patients’ districts for M-MC. Of note, all MC were within less than 100 km from one another.

ILD diagnoses varied across centers (Figure 2). We report an unusually high frequency of idiopathic pulmonary fibrosis (IPF, 58%) in the peripheral C-MC, significantly higher than in the other three central MCs (13–31%). IPF was also the leading diagnosis in S-MC (31%), while in TA-MC and M-MCs the most common diagnosis was autoimmune-related ILD (27% and 30%, respectively). HP was frequently diagnosed in TA-MC (18% of all ILDs), while in the other three centers HP represented a significantly lower proportion of all ILDs, in the range of 1–9%. The most common autoimmune diseases in the autoimmune-related ILD category were rheumatoid arthritis (range 19–47%), systemic sclerosis (10–28%), and myositis/anti-synthetase syndrome (12–28%) (Figure 3).

Diagnosis distribution according to sex is shown in Appendix A. The most common diagnosis for women was autoimmune-related ILD, except in C-MC where it was the second most common after IPF. However, in all four centers, the most common diagnosis for men was IPF. Interestingly, occupational-related ILD diagnosis was assigned exclusively to men.

### 3.1. Patient Characteristics

Baseline characteristics for each center are presented in Table 1. For this analysis, we excluded the 54 subjects with sarcoidosis. Overall, mean age was 67 ± 13, 60% were males, and 57% were past or current smokers. The majority of patients at all four centers had at least one comorbidity (range 67–91%), the most common being arterial hypertension and dyslipidemia.

There was a wide variability in subjects’ characteristics between centers. Males were the majority in most centers (ranges 56–68%), except for TA-MC (50%). Mean age was different across centers, ranging from 63 to 71 years (*p* < 0.01), and was highest at C-MC. Age and smoking distribution by sex also varied between centers, as depicted in Appendix A, respectively. The mean number of pack-years differed between centers, being highest in C-MC (Table 1). In addition to older age and increased smoking, other notable differences were observed in the peripheral C-MC compared to other centers in central Israel. C-MC had the lowest rate of non-ILD pulmonary comorbidities (9%, *p* < 0.01), while it had the highest rates of most cardiovascular risk factors, such as hypertension (48%, *p* < 0.01), dyslipidemia (47%, *p* = 0.01), and cardiovascular disease (37%, *p* < 0.01). Forced vital capacity (FVC) and diffusion capacity for carbon monoxide (DLCO) were highest in C-MC (*p* < 0.01 for both).

Most of the patients ever worked in at least one job that involved inhalational exposure to substances (Figure 4), while the percentage of patients who only worked in office/sedentary jobs involving no relevant occupational exposures was remarkably similar across MCs (range 39–44%). Of note, there was a relatively high frequency of patients who worked as a seamstress or in a job with exposure to fabrics, which was the single most frequent exposure category in TA-MC and M-MC. Other leading categories are construction, drivers, marble industry, army/police and medicine.

### 3.2. Periphery vs. Center of Israel

An important focus of this study was to explore differences in ILD between the periphery and the center of Israel, and identify potential disparities. Baseline characteristics of ILD patients in central compared to peripheral areas of Israel are presented in Appendix A. In the periphery, diagnosis of IPF was more frequent (53% vs. 24%, *p* < 0.01), and classical risk factors for IPF were also increased (older age, male sex, and higher pack-years smoking). On the other hand, exposure-related ILD (5% vs. 16%, *p* < 0.01, mostly driven by HP) and autoimmune-related ILD (16% vs. 25%, *p* < 0.01) were more frequent in central Israel.

To better understand the differences between central and peripheral Israel, we analyzed the predictors for main ILD diagnoses according to patients’ area of residence, as appears in Table 2. IPF was associated with male sex, older age, and higher pack-years in both peripheral and central Israel. History of ever-smoking or cardiovascular disease were more prevalent in subjects with IPF, although they were statistically significant only among those living in peripheral areas (*p* < 0.01 and *p* = 0.03, respectively). Subjects with exposure-related ILD had a younger age in both areas of residence, although it was non-significant among those living in the center (*p* = 0.07). Autoimmune-related ILD was associated with female sex, younger age, and no smoking history in both peripheral and central Israel (*p* < 0.01).

Next, we evaluated the region of residence as a predictor of each main ILD diagnosis. Living in the periphery was associated with IPF (OR 3.38, 95% CI 2.5–4.6, *p* < 0.01). Living in the center was associated with autoimmune ILD (OR 0.53, 95% CI 0.37–0.76, *p* < 0.01) and with exposure-related ILD (OR 0.24, 95% CI 0.14–0.40, *p* < 0.01). Multivariate analysis to assess for independent predictors to each ILD diagnosis is shown in Table 3. Living in a peripheral area was an independent predictor for IPF (adjusted OR 2.95, 95% CI 2.1–4.1, *p* < 0.01). Living in a central area was an independent predictor for exposure-related ILD (AOR 0.46, 95% CI 0.33–0.63, *p* < 0.01). The region of residence did not remain a significant predictor for autoimmune-related ILD in multivariable analysis (AOR 0.70, 95% CI 0.48–1.05, *p* = 0.09).

## 4. Discussion

This is the first description of the characteristics of ILD patients in Israel, including 927 patients from four large tertiary medical centers. These four centers service four main districts (Dan, Center, Haifa, Northern) that are home to the majority (67.8%) of Israel’s population according to the 2021 report by the Israeli Central Bureau of Statistics [15]; therefore, we believe that this data is a good representation of ILDs in Israel. Israel has unique characteristics, including a small geographic size on one hand and a heterogenic population on the other hand, and previous national reports of other lung diseases yielded valuable information [16,17].

We found that the distribution of ILD diagnoses were different across centers, even ones in close geographic proximity. In C-MC, located in Haifa and servicing the peripheral Haifa and Northern districts, we report an unusually high frequency of patients with IPF (58% of all ILD patients). This may be partially explained by the fact that several known risk factors for IPF (age, male sex, and mean pack-years smoking) were highest in individuals treated at C-MC. Of these, smoking is a modifiable risk factor, with health policy implications that will be further discussed below. Additional explanations may include a high exposure to air pollution in the Haifa bay area [18,19,20], and a higher representation of Arab population in C-MC which may have different environmental and occupational exposures [21,22,23]. Indeed, Arabs were over-represented in the IPF group (23.9% compared to 16.9% in non-IPF ILD), although this difference did not reach statistical significance level. The ILD clinics in M-MC and TA-MC have close collaborations with their rheumatology units, which may result in higher rates of referral of autoimmune patients to the ILD clinics, potentially explaining the high rate of autoimmune-related ILDs in these centers, surpassing even IPF.

To further evaluate the reasons for the differences in ILD diagnoses between peripheral and central Israel, we conducted a univariate analysis. We show that the main risk factors for each disease were similar between the central and peripheral areas. These traditional risk factors, such as older age and male sex for IPF, are in line with previous literature [24]. While it validates the diagnoses made between the centers, it also highlights that additional factors might be the cause for this variability, as discussed above.

Most of the patients worked during their lifetime in at least one job that involved inhalational exposure to potentially relevant substances. This is higher than previous reports in the literature [25,26,27]. Nevertheless, only 1–5% of the patients in all four centers were diagnosed with actual occupational-related ILD, meaning that most exposures were not considered the direct cause of the disease, but may have still contributed to the disease risk [28]. Interestingly, we found that there is a relatively high percentage of patients who worked as seamstresses or in a job with exposure to fabrics (3–9%), which is a reported yet less recognized association with ILDs [29,30]. Another common occupation among ILD patients in Israel was professional drivers (3–7%, including taxi, bus, and truck drivers), which is also not well recognized as a risk factor. It is possible that exposure to smoke and gas fumes, along with a higher propensity of smoking in this population, contribute to ILD risk in drivers [31,32,33]. Unfortunately, in our retrospective registries we did not have in-depth data on specific environmental or occupational exposures such as dust, smoke or fumes.

TA-MC was the only center that conducted a prospective registry and routinely used the CHEST ILD Questionnaire, a commonly used patient-reported questionnaire exploring ILD-related symptoms, smoking history, family history, exposures, occupational history, comorbidities, and medication history [14,34]. These may explain the higher detection rate of occupational and environmental exposures in the TA-MC registry. The prospective nature and CHEST ILD Questionnaire likely contributed to the higher percentage of HP diagnosis (18%) in TA-MC compared to 1–9% in the other centers. HP may have been underdiagnosed in these centers since it is known from other registries in the world that HP is a relatively common ILD [35,36].

Several health policy implications could be drawn from our results. We report an unusually high frequency of patients with IPF in the periphery of Israel compared to the center. We show that ever-smoking is a significant risk factor for IPF in the periphery, and that the percentage of Arabs is higher in peripheral C-MC compared to central TA-MC and S-MC, and in IPF compared to non-IPF ILD. The 2020 Israel Minister of Health’s report on smoking in Israel indicates higher smoking rates and quantities in Arab compared to Jewish population [37]. In addition, smoking and smoking-related diseases such as COPD are known to be more frequent in rural compared to urban populations in the US [38,39]. Therefore, it is important to reduce smoking-related disparities by strengthening smoking cessation programs in the periphery in general, and in the Arab population in particular. This goal can be enhanced by government funding for culturally and linguistically adapted digital tools (such as tailored Web-based intervention program and applications) [40], and training health care professionals to take an individualized approach according to cultural and socioeconomic background [41]. Importantly, these measures will also help prevent other smoking-related diseases such as COPD and lung cancer.

In addition to smoking, other factors may account for the higher frequency of IPF in peripheral areas, including occupational and environmental. Exposures are important in other ILDs as well, and in particular in HP. We recommend a thorough and systematic evaluation of occupational and environmental exposures in all ILD clinics, which may be facilitated by a structured, locally adapted patient questionnaire (e.g., the CHEST ILD questionnaire [14]). These data could improve exposure detection and prevention on the patient-level, and could be aggregated to monitor deviations of exposures and ILD morbidity on the regional and national levels. Israel currently lacks a centralized, national respiratory exposure registry, similar to that of the US National Institute for Occupational Safety and Health (NIOSH) Respiratory Health Program. We recommend establishing one to enable early detection of clusters of exposure-related ILDs. A striking example for the need to have a national exposure registry are the multiple cases of artificial-stone silicosis that were discovered late and resulted in lung transplantation [42], many of which could have been detected early or even prevented with such a national program.

We are aware that this study has limitations. Among them, three registries were retrospective compared to one that was prospective. Retrospective analysis, although performed by reviewing all clinic visits and investigations, could result in failure to capture non-documented data and to perform predefined structured analysis of each case. In addition, as stated above, the use of a specific questionnaire to screen for relevant exposures might have also contributed to the differences in characteristics and diagnoses between centers. The C-MC and M-MC registries did not include sarcoidosis patients, while the other ones did, therefore we had to exclude sarcoidosis from the comparative analysis. We could not include all tertiary MCs in Israel due to lack of ILD registry data. However, our four MCs service four large districts in Israel, which are home to the majority of the Israeli population, and are among the largest referral centers for ILD in the country. Unfortunately, the Southern, Jerusalem, and Judea and Samaria districts are almost not represented in our data, although these districts account for a significant minority of the Israeli population. In addition, patients that are being treated in primary or secondary care settings are not represented, leading to a possible selection bias of more severe cases. A prospective nationwide Israeli ILD registry was recently launched and will hopefully provide these missing data.

## 5. Conclusions

There were significant variations in ILD patient characteristics among different centers in Israel, including sex, age, smoking, exposure history, and ILD diagnoses, despite close geographic proximity between centers. IPF and autoimmune ILDs were the two most common etiologies, and HP may have been underdiagnosed in some centers. We report an unusually high frequency of IPF in the northern peripheral C-MC, that is likely the result of potential disparities that should be addressed. The next step is conducting a prospective national ILD registry in order to standardize the data collection, eliminate biases, and lead to further insights that may have actionable implications.

## Figures and Tables

**Figure 1 diagnostics-15-02780-f001:**
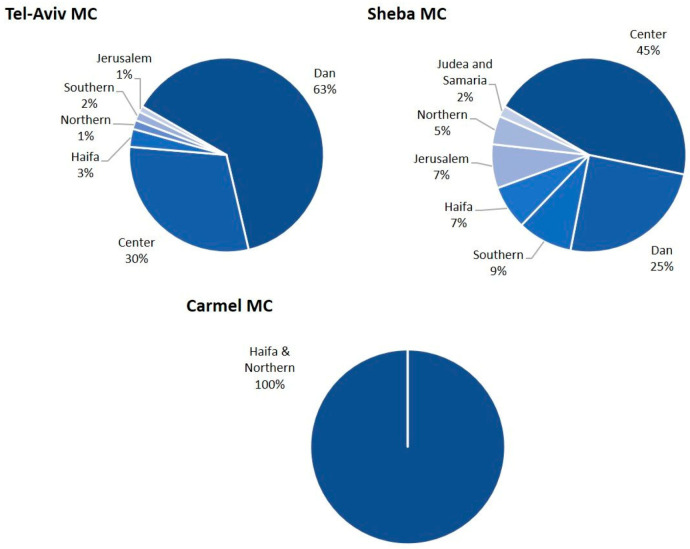
Patient distribution according to district. Dan, Center, and Jerusalem districts are considered central Israel, while all other districts represent the periphery. Abbreviations: MC, medical center.

**Figure 2 diagnostics-15-02780-f002:**
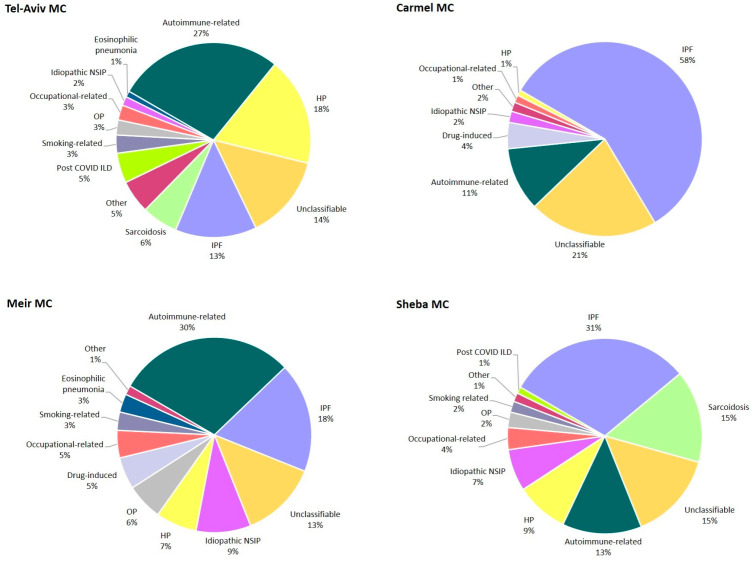
Interstitial lung disease diagnosis distribution by medical center. Carmel and Meir registries did not include patients diagnosed with sarcoidosis. Smoking-related ILD category includes RB-ILD, PLCH and DIP. Abbreviations: DIP, desquamative interstitial pneumonia; HP, hypersensitivity pneumonitis; ILD, interstitial lung diseases; IPF, idiopathic pulmonary fibrosis; NSIP, non-specific interstitial pneumonia; OP, organizing pneumonia; PLCH, pulmonary Langerhans cell histiocytosis; RB-ILD, Respiratory bronchiolitis-interstitial lung disease.

**Figure 3 diagnostics-15-02780-f003:**
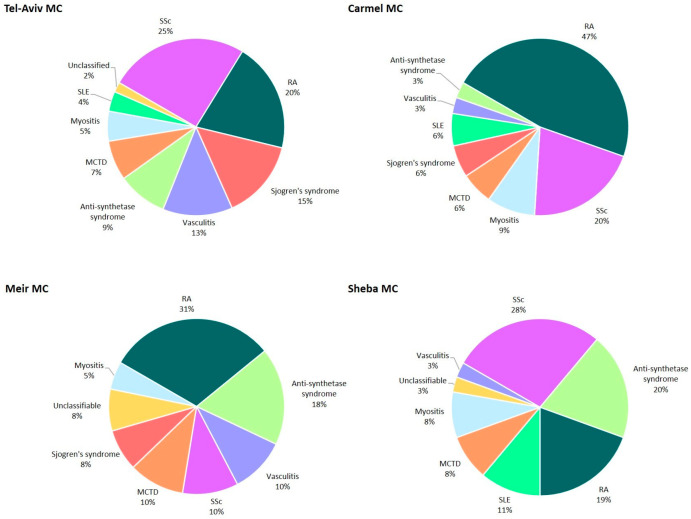
Breakdown of autoimmune diagnoses in each medical center. Abbreviations: MCTD, mixed connective tissue disease; RA, rheumatoid arthritis; SLE, systemic lupus erythematosus; SSc, systemic sclerosis.

**Figure 4 diagnostics-15-02780-f004:**
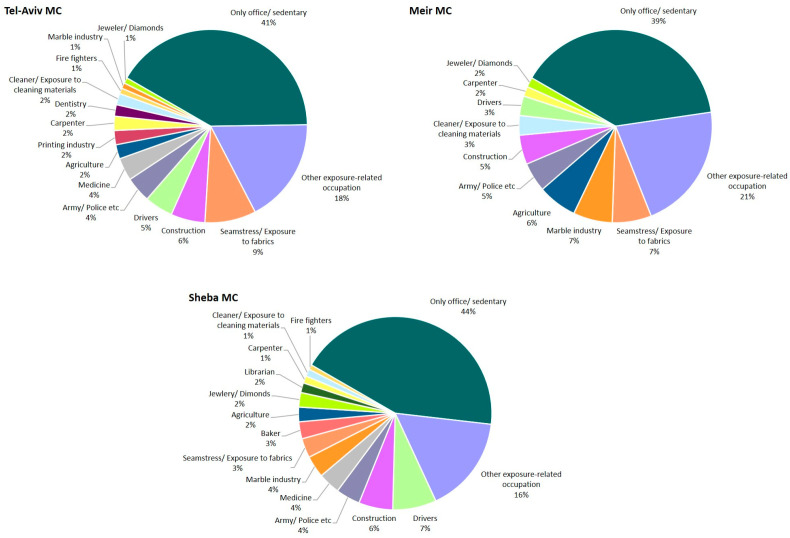
Occupation distribution in each medical center. Patients could be included in more than one exposure-related occupation based on prior exposures.

**Table 1 diagnostics-15-02780-t001:** Baseline characteristics by medical center, excluding sarcoidosis patients.

Variable	S-MC	TA-MC	M-MC	C-MC	Total	*p*
	N = 232 (%)	N = 188 (%)	N = 132 (%)	N = 321 (%)	N = 873 (%)	
Female sex	96 (41)	94 (50)	58 (44)	102 (32)	350 (40)	<0.01
Age, mean (SD) ^#^	63 (13)	68 (12)	65 (14)	71 (11)	67 (13)	<0.01
Peripheral residence	54 (23)	12 (6)	N/A	321 (100)	387 (52)	<0.01
Ever smoker	126 (54)	110 (59)	68 (51)	189 (59)	493 (57)	0.42
Pack years, median (IQR)	30 (15–50)	26 (11–50)	30 (20–40)	35 (20–50)	30 (18–50)	0.07
EO exposures ^^^	121 (52)	145 (77)	69 (52)	N/A	335 (61)	<0.01
Ethnicity						
Jews	162 (82.2)	182 (96.8)	N/A	242 (75.3)	321 (43)	<0.01
Arabs	28 (14.2)	4 (2.1)	N/A	79 (24.7)		
Other	7 (3.6) ^¶^	2 (1.1)	N/A	0		
Comorbidities						
Non-ILD lung disease	33 (14)	42 (22)	17 (13)	28 (9)	120 (14)	<0.01
Hypertension	67 (29)	87 (46)	53 (40)	153 (48)	360 (41)	<0.01
Diabetes	56 (24)	52 (28)	31 (24)	85 (27)	224 (26)	0.77
Dyslipidemia	84 (36)	84 (45)	43 (33)	151 (47)	362 (42)	0.01
Cardiovascular disease	51 (22)	51 (27)	32 (24)	118 (37)	252 (29)	<0.01
Heart failure	26 (11)	26 (14)	12 (9)	49 (15)	113 (13)	0.26
History of Cancer	42 (18)	31 (17)	13 (10)	51 (16)	137 (16)	0.21
Pulmonary functions, mean (SD) ^#^						
FVC (% pred) ^¶^	72.5 (23)	76.8 (19)	75.7 (20)	81.2 (21)	77.1 (22)	<0.01
DLCO (% pred) ^¶^	59.1 (19)	53.6 (19)	51.7 (18)	60.9 (19)	54.7 (20)	<0.01

Abbreviations: MC, medical center; S-MC, Sheba medical center; TA-MC, Tel-Aviv medical center; M-MC, Meir medical center; C-MC, Carmel medical center; DLCO, diffusion capacity for carbon monoxide; EO, environmental and occupational; FVC, forced vital capacity; ILD, interstitial lung disease; SD, standard deviation; IQR. Interquartile rang; N/A, not available. ^#^ Age and PFTs were obtained at presentation. ^^^ Patients with any environmental or occupational exposure known in the medical literature to be related to ILD, whether or not being the specific cause for a patient’s ILD. ^¶^ Missing data for FVC—95 subjects, for DLCO—193 subjects, for ethnicity at S-MC—35 subjects.

**Table 2 diagnostics-15-02780-t002:** Predictors for main interstitial lung diseases in central and peripheral Israel.

ILD	Variable	Central		Peripheral	
		OR (95% CI)	*p*	OR (95% CI)	*p*
Idiopathic pulmonary fibrosis	Older age	1.05 (1.03–1.07)	<0.01	1.04 (1.02–1.06)	<0.01
	Female sex	0.44 (0.27–0.73)	<0.01	0.24 (0.15–0.38)	<0.01
	Ever smoking	1.56 (0.95–2.38)	0.09	2.22 (1.49–3.35)	<0.01
	Pack years	1.01 (1.00–1.02)	0.04	1.01 (1.00–1.02)	0.04
	CVD	1.48 (0.89–2.51)	0.14	1.61 (1.05–2.47)	0.03
Autoimmune-related	Older age	0.98 (0.96–0.99)	0.01	0.95 (0.93–0.97)	<0.01
	Female sex	4.13 (2.43–6.97)	<0.01	4.23 (2.36–7.49)	<0.01
	Ever smoking	0.61 (0.38–0.99)	0.04	0.50 (0.29–0.88)	0.03
Exposure related ^^^	Older age	0.98 (0.96–1.00)	0.07	0.94 (0.87–0.94)	<0.01
	Female sex	1.34 (0.79–2.30)	0.36	0.74 (0.26–2.10)	0.52
	Ever smoker	0.68 (0.39–1.16)	0.16	0.67 (0.27–1.68)	0.41

Abbreviations: CVD, cardiovascular diseases; ILD, interstitial lung disease; OR, odds ratio. ^^^ Including hypersensitivity pneumonitis and cases of pneumoconiosis.

**Table 3 diagnostics-15-02780-t003:** Peripheral residence as an independent risk factor for ILD subtype.

Interstitial Lung Disease	Living in Peripheral vs. Central Area		
	Adjusted OR	95% CI	*p*-Value
Idiopathic pulmonary fibrosis ^#^	2.95	2.12–4.11	<0.01
Autoimmune-related	0.7	0.48–1.05	0.09
Exposure-related ^^^	0.46	0.33–0.63	<0.01

Abbreviations: ILD, interstitial lung disease; OR, odds ratio. ^#^ IPF was also adjusted for the presence of cardiovascular disease. ^^^ Including hypersensitivity pneumonitis and cases of pneumoconiosis.

## Data Availability

The raw data supporting the conclusions of this article will be made available by the authors on request.

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
