# Peer review of "Interstitial Lung Diseases in Israel: Large Variability in Close Geographic Proximity"

_diagnostics, 2025, doi:10.3390/diagnostics15212780_

Round 1

Reviewer 1 Report

Comments and Suggestions for Authors

The topic is current and interesting. I have some comments:

1- Abstract. The study is based on data from only four tertiary centers in Israel, which may not represent the entire population, especially patients managed in primary or secondary care settings. This could limit the generalizability of the findings to the broader Israeli population. Do you have some comments?

2. Abstract. While the study mentions occupational exposures, it does not provide in-depth analysis of specific environmental or occupational factors that could influence ILD development. This limits the ability to draw definitive conclusions about exposure-related risks across different regions. Please, I suggest to comment this observation.

3. 1. Introduction 45-50. Interstitial lung disease (ILD) is a broad term encompassing over 200 fibrotic and  non-fibrotic disorders that diffusely affect lung parenchyma. Typically, ILD cause lung  restriction and impaired gas exchange, resulting in breathlessness, diminished exercise  
tolerance, reduced quality of life, and in some cases, shortened survival [1]. Diagnosis of  
a specific ILD relies on the combination of clinical, radiological, and pathological charac-
teristics, preferably through a multidisciplinary discussion (MDD) [2,3]. Authors are kindly requested to emphasize the current concepts about these issues in the context of recent knowledge and the available literature. These articles should be quoted in the References list. References 1. Viewpoint: a multidisciplinary approach to the assessment of patients with systemic sclerosis-associated interstitial lung disease. Clin Rheumatol. 2023 Mar;42(3):653-661. doi: 10.1007/s10067-022-06408-4.;   2. High-Resolution Computed Tomography: Lights and Shadows in Improving Care for SSc-ILD Patients. Diagnostics (Basel). 2021;11(11):1960. doi: 10.3390/diagnostics11111960.

4. Methods/Results. I suggest authors to include more detailed information on the validation and standardization of data collection methods across the different centers to ensure consistency and comparability of the data.

5. Methods/Results. I suggest authors to provide a clearer explanation of how missing data were handled in the analysis to improve transparency and reproducibility. 

6. Discussion/Conclusions. Enhance the clarity of comparisons between centers by providing more detailed explanations of the observed differences, ensuring that the reasoning behind these variations is explicitly articulated. Please, add some comments.

7. Discussion/Conclusions. Increase transparency regarding the limitations of retrospective data, explicitly addressing potential confounding factors and biases that may affect the interpretation of the results. I suggest to improve this part.

Author Response

Please see the attached form

Reviewer 2 Report

Comments and Suggestions for Authors

Thank you for the opportunity to review your manuscript. 

The main question addressed by the research is to provide a comprehensive report on (mainly fibrotic) interstitial lung diseases in Israel. As this comprises a specific population, and there are no other reports on this population available yet, it does provide an original contribution to the field. It furthermore illustrates that the prevelance of different types of ILD does vary by region, and might be influenced by different population make up or different exposures.

With regard to the methodology, the authors could add more information about the background of selecting participating hospitals in this study. Do the four Tertiary Hospitals referenced to in this article represent all Tertiary Hospitals for ILD in Israel?

The conclusions do seem to be consistent with the evidence presented, and the authors appropriately acknowledge the limitations to their research. The references also seem to be appropriate.

Author Response

Please see the attached form

Reviewer 3 Report

Comments and Suggestions for Authors

The retrospective epidemiological study reports data on interstitial lung diseases in Israel.
The topic is interesting primarily because it provides more accurate data and can be considered an initial registry of pulmonary fibrosis in this country.
The accuracy of the methodology, the tables, and the statistical analysis are highly valid and allow for an easier reading of the article.
I consider this study a very interesting first step towards better understanding interstitial lung diseases in that region and facilitating the understanding and study of the disease from both a clinical and research perspective.

Author Response

Please see the attached form

Round 2

Reviewer 1 Report

Comments and Suggestions for Authors I have carefully evaluated the revised manuscript. Tha manuscript has been improved. I have no further comments.  
